# Eliciting Preferences of Providers in Primary Care Settings for Post Hospital Discharge Patient Follow-Up

**DOI:** 10.3390/ijerph18168317

**Published:** 2021-08-05

**Authors:** Xin Wang, Kuimeng Song, Lijin Chen, Yixiang Huang, Stephen Birch

**Affiliations:** 1School of Public Health, Sun Yat-Sen University, Guangzhou 510080, China; wangxin25@mail.sysu.edu.cn (X.W.); chenlijin@mail.sysu.edu.cn (L.C.); 2Shandong Institute of Medicine and Health Information, Shandong First Medical University & Shandong Academy of Medical Science, Jinan 250012, China; kmsong@sdfmu.edu.cn; 3Centre for the Business and Economics of Health, University of Queensland, Brisbane, QLD 4072, Australia; stephen.birch@uq.edu.au

**Keywords:** integrated care, post-hospital discharge follow-up, health care provider, preference

## Abstract

Background: Post-hospital discharge follow-up has been a principal intervention in addressing gaps in care pathways. However, evidence about the willingness of primary care providers to deliver post-discharge follow-up care is lacking. This study aims to assess primary care providers’ preferences for delivering post-discharge follow-up care for patients with chronic diseases. Methods: An online questionnaire survey of 623 primary care providers who work in a hospital group of southeast China. Face-to-face interviews with 16 of the participants. A discrete choice experiment was developed to elicit preferences of primary care providers for post-hospital discharge patient follow-up based on six attributes: team composition, workload, visit pattern, adherence of patients, incentive mechanism, and payment. A conditional logit model was used to estimate preferences, willingness-to-pay was modelled, a covariate-adjusted analysis was conducted to identify characteristics related to preferences, 16 interviews were conducted to explore reasons for participants’ choices. Results: 623 participants completed the discrete choice experiment (response rate 86.4%, aged 33 years on average, 69.5% female). Composition of the follow-up team and adherence of patients were the attributes of greatest relative importance with workload and incentives being less important. Participants were indifferent to follow-up provided by home visit or as an outpatient visit. Conclusion: Primary care providers placed the most importance on the multidisciplinary composition of the follow-up team. The preference heterogeneity observed among primary care providers suggests personalized management is important in the multidisciplinary teams, especially for those providers with relatively low educational attainment and less work experience. Future research and policies should work towards innovations to improve patients’ engagement in primary care settings.

## 1. Introduction

Chronic diseases have been the predominant challenge worldwide as a result of the epidemiological transition, rapid population aging, and fragmented healthcare systems [1,2]. Integrated care is suggested as one strategy to improve the continuity, efficiency, and quality of care for patients with chronic conditions [3,4]. Follow-up of patients with chronic diseases following hospital discharge could promote collaboration among different providers and continuity in care processes [5,6]. It has been demonstrated that patients who receive early follow-up post hospital discharge have fewer unplanned hospital readmissions, emergency department visits, and complications, higher quality of life, and lower annual health care expenditures [6,7,8,9]. 

China has developed and implemented people-centered integrated healthcare systems in the last decade [10,11]. In a rural county or urban district, medical consortia are formed to organize integrated care delivery by collaboration across all health-related institutions within the county/district. By December 2019, there had been 1408 and 3346 medical consortia developed in urban districts and rural counties in China. Follow-up of patients with diabetes and hypertension has been mandatory under the Equalization of Essential Public Health Service (EEPHS) program nationwide since 2009 [12]. According to the EEPHS program guidelines, medical staff in primary care settings are responsible for four face-to-face follow-ups, measuring blood pressure, weight, and heart rate, guiding medication and healthy lifestyle, and referring patients to other providers if necessary. However, follow-up of patients with chronic diseases is not mandatory under the policy. Some medical consortia have started following up discharged patients to reduce unplanned readmissions and health care expenditure, motivated by the incentives under the “Global budget, balance retained” arrangement [13]. However, there is an absence of guidelines for post-discharge follow-up.

Participants’ perspectives should be considered in designing follow-up programs for discharged patients, in order to improve feasibility, acceptability, and effectiveness. Some studies have explored patients’ preferences for cancer post discharge follow-up care using discrete choice experiments (DCEs) or best-worst scaling (BWS), with inconsistent results. Gulliford et al. compared experiences of 193 UK patients with breast cancer, who were randomized to receive either conventional intensive follow-up or less intensive follow-up, to find that women appeared to prefer non-intensive follow-up schedules [14]. Another study in the United Kingdom found that most patients felt their follow-up to be too frequent and were in favor of a less-intensive, symptom-driven follow-up [15]. In contrast, two binary DCE studies and a BWS study exploring preferences for breast, and head and neck cancer follow-up services in the Netherlands, Australia, and Italy all concluded that patients were more likely to choose an intensive follow-up scheme with face-to-face contacts and intensive visits (every 3–6 months). It should be noted that the definition of intensive and non-intensive follow-up is inconsistent among different healthcare systems and during different time periods. Predictive factors and their impact on the rate of compliance with follow-up in discharged patients were determined by correlation and logistic regression analysis [16,17]. It has been illustrated that male, marital status, higher education, and low incidence of comorbidities were associated with lower loss to follow-up after discharge. Few studies considered post-discharge follow up from the perspective of providers. Heather et al. provided insights into different types of oncologists’ perspectives of their roles and responsibilities during multi-disciplinary breast cancer follow-up using qualitative content analysis [18]. To our knowledge, there is no stated preference survey of providers’ perceptions for follow-up of discharged patients. 

We aimed to elicit preferences of providers in primary care settings for follow-up of discharged patients using a discrete choice experiment. Moreover, as socio-demographic or work-related characteristics may have some effect on providers’ preferences, we further analyzed the influence of these characteristics over preferences for different follow-up team compositions in order to provide insight for policy making in the 4000+ district/county medical consortia about follow-up of discharged patients in integrated primary care settings of China and other countries.

## 2. Materials and Methods

### 2.1. Study Setting and Sampling

Yangxi County is located on the southwest coast of Guangdong Province, China. In order to promote integrated care, Yangxi Hospital Group (YHG) was established in 2017, consisting of three county-level hospitals and eight township health centers (THCs). YHG is planning to provide follow-up care for discharged patients with diabetes and hypertension to promote the integration of care. Based on the EEPHS program, all patients with diabetes and hypertension in Yangxi county are managed in the Integrated Care System for patients with Chronic Diseases (ICSCD). According to the existing implementation plan, once a patient with diabetes or hypertension is discharged from one of the three county-level hospitals, the patient is placed on the follow-up list of the THC where the patient lives. The patient’s healthcare record and sociodemographic information is shared among providers in hospitals and THCs and the patient is followed-up within seven days of discharge. Data collected at the follow-up is shared in the ICSCD, and follow-ups are repeated every week, four weeks, or twelve weeks according to risk status of the patient. However, there is little evidence available on motivating primary care providers to participate in the follow-ups.

Cluster sampling was used in this study. All physicians, nurses and other providers working in the eight THCs of YHG were invited to participate in a web-survey. The survey was conducted in November 2020, after China managed to control COVID19. Participation in the survey was voluntary with a consent form. Participants were not compensated. In addition, 2 providers in each THC were purposively sampled and interviewed about reasons underlying the choices made. This study (No. 2020-114) was approved by the Ethical Commission of School of Public Health, Sun Yat-Sen University. 

### 2.2. DCE Design

In this study, the attributes and levels for the DCE were chosen based on literature reviews, key informant interviews (*n* = 8) with researchers working on primary care and a focus group discussion with 2 physicians, 2 nurses, and 2 public health physicians from three THCs in YHG. Eventually, six attributes, including team composition, workload, follow-up pattern, adherence of patients, incentives, and payments, were included in this study. Table 1 lists the attributes and corresponding levels.

A full factorial design produced 3^6^ = 729 scenarios. An orthogonal factorial experimental design was then used to generate a more manageable level of 18 scenarios. One job scenario with “middling” attributes was chosen as a constant alternative, and the other 17 alternatives compared with it. Previous studies have suggested this approach could make the choices easier to understand [19]. To avoid overloading the respondents, the 17 pairs of choices were split randomly across two versions of the questionnaires, one version with 9 choices and the other version with 8 choices. The two versions of the questionnaires were then randomly allocated to the respondents.

### 2.3. Conceptual Framework and Data Analysis

The conceptual framework of this study is based on Lancaster’s theory, which assumes that the utility associated with a good or service is made up of the utilities of its composing characteristics [20]. The utility acquired by individual n from alternative j can be decomposed into an explainable or systematic component $V_{nj\;}$ and a non-explainable or random component $\varepsilon_{nj}$:$$U_{nj}=V_{nj}+\varepsilon_{nj}\;\;\;\;\;\;j=1,\ldots ,J$$

In this study, the systematic component $V_{nj}$ is a function of observed attributes of the follow-up groups $X_{nj}$ (the six attributes included in the study, including team composition, workload, etc) and observed characteristics of health workers $Z_{n\;}\;$(demographic characteristics of the respondents, including gender, age, etc), while the random component $\varepsilon_{nj}$ is related to unobserved attributes or preference variation:$$V_{nj}={\beta x}_{nj}+{\gamma z}_{n}\;\;\;\;\;\;j=1,\ldots ,J$$
where β and y are vectors of coefficients to be estimated.

Random utility theory assumes that the individual acts rationally and chooses the alternative with the highest level of utility. When individual n is asked to choose between two alternatives, i and j, then the probability that alternative i is chosen is given by:$$\begin{array}{cc} P_{ni} & =\mathrm{Prob}(U_{ni}>U_{nj})\\ & =\mathrm{Prob}(V_{ni}+\varepsilon_{ni}>V_{nj}+\varepsilon_{nj})\\ & =\mathrm{Prob}(V_{ni}-V_{nj}>\varepsilon_{nj}-\varepsilon_{ni})\;\forall i\neq j \end{array}$$

When the random component $\varepsilon_{nj}$ is assumed to be independent and identically distributed extreme value, the probability that individual n chooses alternative j can be estimated with a conditional logit model:$$P_{ni}=\frac{e^{V_{ni}}}{\sum_{j}e^{V_{nj}}}\;\;\;\;\;\;j=1,\ldots ,J$$

The estimated coefficients in the regression give information about the direction and significance of the effects of changing the levels of one attribute, but it does not provide a valuation necessary for comparison with alternative policies. We therefore calculate willingness to pay (WTP), based on regression results using Stata’s nlcom command. In this study, the WTP for a higher level of a specific follow-up group attribute x is measured by how much the providers would be willing to forgo in payments in order to have their preferred follow-up protocol adopted. As the payment per follow-up visit is continuous, the WTP for attribute can be estimated as:$$\mathrm{WTP}\left(x\right)=-\frac{\partial U/\partial x}{\partial U/\partial payment}$$

All statistical analyses were performed using STATA 14.0. Interviews with 16 providers were transcribed word by word, and thematic analysis was conducted by MAXQDA 11.

## 3. Results

### 3.1. Respondents’ Characteristics

All health workers in the eight THCs of YHG were invited to participate in the web-survey and 623 agreed to participate (response rate 86.40%). Table 2 shows the demographic characteristics of the sample. There were 159 physicians, 243 nurses, and 221 other providers (including public health physicians, pharmacists, and laboratory technicians). Overall, most respondents were female (69.50%), married (66.61%), and the mean age was 33. Most of the respondents had a relatively low education status and professional title (only 14% had a bachelor or higher degree) and held an intermediate or senior professional title. Most respondents had more than 5 years of work experience (64.69%) and a monthly income between 3000 and 5000 RMB (51.85%).

Table 3 also shows the monetary valuation (willingness to pay) of each attribute and level. A positive result indicates how much providers need to be compensated per follow-up visit to accept that specific attribute or level, and a negative result indicates how much payment per follow-up visit providers would be willing to give up in order to get that attribute or level. Compared to the reference levels, the amount of money that providers in THCs would like to forgo in return for certain attributes and levels are as follows: team composition, advanced (¥2.282); team composition, regular (¥1.673); adherence of patients, good (¥1.442); incentive, i.e., rewards for doing well, penalties for not doing well (¥0.899), and adherence of patients, very good (¥0.759). Additionally, compared to serving no more than 20 patients, health workers would accept additional compensation of ¥1.310 RMB per follow-up visit in order to serve 40–59 patients.

### 3.2. Analysis with Interaction Terms

Table 4 shows the results of the influence of socio-demographic or work-related characteristics over preferences among different follow-up team choices, according to a stepwise conditional logit model. Only those interaction terms with statistical significance remained in the final model. Income interaction terms (<¥3000; ¥3000~¥5000; ≥¥5000) were not statistically significant, therefore they were excluded in the stepwise logit model. Firstly, compared with male health workers, female health workers valued multidisciplinary teams less. Secondly, health workers with different education status differed in their preferences for team composition. Compared to health workers with associate degrees, health workers with bachelor degrees had a stronger preference for a follow-up team with multidisciplinary composition and with home visit as the follow-up mode, while health workers with technical school education placed a lower value on incentive mechanisms with penalties. Lastly, health workers with more work experience placed a lower value on good adherence of patients.

## 4. Discussion

Since diabetes and hypertension may lead to vascular complications, which are major causes of mortality and morbidity in the long term, appropriate follow-up for discharged patients is critical to reduce complications and increase patient quality of life [21,22]. Taking multiple providers’ perceptions into consideration is essential to improve the quality and efficiency of follow-up [23].

### 4.1. Main Findings

The results suggest that providers in primary care settings preferred a follow-up team with higher payments, multidisciplinary team members, less workload, incentive mechanisms without penalties, and good adherence of patients. They prioritized composition of the follow-up team when deciding which team to take part in. This was further supported by the results of willingness to pay analysis and face-to-face interviews with health workers. Health workers were willing to give up ¥1.673 or ¥2.282 of payment per follow-up visit respectively to join a regular team (GP, nurse, public health physician) or an advanced team (GP, nurse, public health physician, pharmacist, and health manager) as compared to a basic team (GP, nurse). Multidisciplinary teams offer several advantages over individual providers. A multidisciplinary team is better placed to meet the physical and mental demands of patients, especially those with multiple chronic diseases and combined medication. In addition, a multidisciplinary team offers a supportive structure in which team members offer each other coaching, encouragement, mentorship, and discipline [24]. Multidisciplinary teams for patients with chronic diseases have been demonstrated to reduce complications and unplanned hospital admissions [25,26]. The COVID-19 pandemic has also highlighted the importance of proactive multidisciplinary team-based care [24]. Although there is no evidence of health care providers’ having a preference for multidisciplinary follow-up teams, some research follow up strategies have been redesigned and implemented s around unmet needs of patients with chronic conditions using the full multidisciplinary teams in developing countries [27,28]. In China, multidisciplinary teams are used for joint treatment for patients with chronic conditions during hospitalization, there is an absence of multidisciplinary teams during post-discharge follow-up, especially in primary health care institutions [29].

Adherence of patients, workload and incentives were of less importance than team composition. Analysis of willingness to pay showed that health workers were willing to give up ¥1.442 of payment per follow-up visit to join a team with good patient adherence and require additional compensation of ¥1.310 per follow-up visit for them to accept serving 20–40 more patients per month. Different from results of research from patients’ perspectives, multi-providers in this study pay no attention to location of the follow-up care [30,31]. There are several possible reasons for this finding. Firstly, where to follow up the patients wasn’t a mandatory requirement in the performance appraisal system. In practice, it usually depends on patient condition and availability. Secondly, it didn’t take more time or effort to provide home visits than outpatient visits, as the providers could integrate the home visits with other routine services for the patients’ family members or neighbors. Regarding the preference for lower workload, research showed that the breadth and volume of the workload of HCPs in primary care institutions had expanded since the new round of healthcare reform in 2009, providing both basic clinical care and public health care [32]. This might be one of the reasons for participants in the study choosing lower workload,

The analysis with interaction terms found significant variation in preferences based on providers’ educational attainment. Providers with high educational attainment place a much higher importance on multidisciplinary team members and home visits. In contrast providers with low educational attainment placed a lower value on a mechanism with penalties. Additionally, gender and working time also have impacts on providers’ preferences. Male participants valued multidisciplinary teams more than female participants, and providers with less work experience considered good patient adherence more important than more experienced providers. This information provides insights on matching team members based on their characteristics.

### 4.2. Policy Implications

Health care systems with high-performing primary care systems achieve better processes of care, better health outcomes and lower overall health care expenditure [33,34]. Primary care follow-up of discharged patients promotes continuity and coordination of health care and improved system performance [35,36]. The principal findings of this study provide evidence for implementing post discharge follow-up in primary practice. 

The first and most important feature of implementing post discharge follow up care is to arrange multidisciplinary follow-up teams with GPs, nurses, public health physicians, pharmacists, health managers, nutritionists, and other health care providers. According to the experience of multi-disciplinary teams (MDTs) in specialist hospitals, joint training and regular or non-regular team meetings promote capacity building and interdisciplinary cooperation in the follow-up teams [37]. Second, follow-up teams’ management is enhanced based on the preferences of different members. Female members benefit from awareness of the importance of multidisciplinary team. Providers with higher educational attainment prefer home visits which contrast with providers with lower educational attainment. Hence, those with lower levels of education might be allocated to patients in low or moderate (rather than high) risks, attending THCs where there is immediate access to other providers when emergency situations arise. Additionally, for providers with low educational attainment and relative low salary, removing disincentives for poor performance would play a big role in motivating them. Providers with shorter work experience, serving patients with better adherence of follow-up care would improve the overall efficiency of serving discharged patients. Last but not least, patients’ awareness of multidisciplinary follow-up teams, integrated care, and follow-up for chronic disease should be improved. Previous research has implied that there is no common recognition of people as participants and not just beneficiaries of health care services among the general population in China [38]. Patients’ higher adherence would lead to more enthusiasm of providers and in turn better patient health outcomes. 

### 4.3. Strengths and Limitations

To our knowledge, this is the first stated preference study to explore primary care providers’ preferences towards follow-up care of discharged patients with chronic diseases. The study design, DCE, quantifies individuals’ priorities by estimating the relative importance of different attributes. Moreover, the results provide evidence of how to implement follow-up teams and how to optimize care processes. This informs policy-making processes concerning integrated care for patients with chronic diseases in the district/county medical consortia in China and health care systems in other countries.

However, this study is subject to some limitations. First, the data were collected through online questionnaires, and the cognitive ability of each participant with respect to completing the task was not evaluated. Participants completing the online questionnaire are likely to be younger, have more education, and possess higher cognitive ability than general primary care providers overall. Although providers participating in the face-to-face interviews showed good cognitive ability about making choices, the web-survey might ignore preferences of older primary care providers. Second, the three levels of the follow-up visit pattern were not highly differentiated. In other research on patient preferences, telephone follow up, home visit, and outpatient visits were compared [31]. We defined the three levels as telephone, telephone and outpatient visit, telephone, and home visit, based on the practical experience of providers. Moreover, the three levels of payment were not associated with the three levels of follow-up pattern. Further research will reveal whether this classification results in providers’ indifference to the follow up visit pattern. Third, considering the broad heterogeneity among provider types, such as physician and nurse, it may be better to analyse their preferences separately. However, we analyzed the preferences of primary care providers as a whole due to the concern that the sample size might not meet the sample size requirements of DCE analysis to perform separate analysis. Last, caution should be taken when the results and policy implications of this study are extended to other primary care systems in China or other countries. The results of this study relate to a combination of healthcare background, human resource management, healthcare delivery status and other factors in Yangxi county. However, the DCE design, rather than the results, is applicable in other primary health care systems.

## 5. Conclusions

Follow-up of patients following hospital discharge is an important part of integrated care for patients with chronic diseases. Our overall results suggest that primary care providers placed the most importance on having a multidisciplinary follow-up team. To further improve efficiency and quality of follow-up care, personalized management is suggested in the multidisciplinary teams, especially for those providers with relatively low educational attainment and less work experience. This study not only makes a contribution to developing policies on implementing post-discharge follow-up, building multidisciplinary teams, and promoting care integration in the over 4000 district/county medical consortia in China, but also provides a methodology for exploring providers’ preferences for models of care delivery in health care systems of other countries.

## Figures and Tables

**Table 1 ijerph-18-08317-t001:** Attributes and levels adopted in the discrete choice experiment.

Attribute	Description	Levels	
Team Composition	…is how many members in your follow-up team and their occupations.	1	Basic team (GP + nurse)
		2	Regular team (GP + nurse + public health physician)
		3	Advanced team (GP + nurse + public health physician + pharmacist + health manager)
Workload	…is how many discharged patients to be followed up by your team per month.	1	1–19 discharged patients
		2	20–39 discharged patients
		3	40–59 discharged patients
Follow-up visit pattern	…is how you and your team conduct the follow-up.	1	Telephone
		2	Telephone + outpatient visits
		3	Telephone + home visit
Adherence of patients	…is how the discharged patients adherent to the follow-up arrangement made by you and your team	1	Not good
		2	Good
		3	Very good
Incentive mechanism	…what kind of incentives provided for team members.	1	No rewards or penalties
		2	No rewards for doing well, but penalties for not doing well
		3	Rewards for doing well, and penalties for not doing well
Payment	…refers to how much received by you for providing follow-up care for discharged patients per time.	1	¥5
		2	¥7
		3	¥9

Notes: GP is short for general physician.

**Table 2 ijerph-18-08317-t002:** Characteristics of the respondents.

	Overall (*n* = 623)	Physicians (*n* = 159)	Nurses (*n* = 243)	Others ^#^ (*n* = 221)	*p*
Gender (%)					
Male	30.50	63.52	3.70	36.20	<0.001
Female	69.50	36.48	96.30	63.80	
Age (mean, SD)	32.97 (9.34)	36.12 (8.59)	28.87 (7.87)	35.21 (9.73)	<0.001
Marital Status (%)					
Married	66.61	78.62	52.26	73.76	<0.001
Not Married	33.39	21.38	47.74	26.24	
Education (%)					
≤Technical School	32.58	8.18	57.20	23.08	<0.001
Associate Degree	53.29	61.64	38.68	63.35	
≥Bachelor Degree	14.13	30.19	4.12	13.57	
Professional Title (%)					
≤Primary	85.71	74.84	87.65	91.40	<0.001
≥Intermediate	14.29	25.16	12.35	8.60	
Work years (%)					
<5 years	35.31	23.27	48.15	29.86	<0.001
≥5 years	64.69	76.73	51.85	70.14	
Monthly Income RMB (%)					
<3000	28.25	13.84	39.09	26.70	<0.001
3000~5000	51.85	55.97	45.68	55.66	
≥5000	19.90	30.19	15.23	17.65	

Notes: ^#^ Others refer to public health physician, pharmacist and laboratory technicians.

**Table 3 ijerph-18-08317-t003:** Model results and willingness to pay (WTP).

Variable ^#^	Coefficient Means	Robust S.E.	Willingness to Pay	95% Confidence Interval
Constant	−0.155	0.128	/	/
Payment	0.169 ***	0.018	/	/
Team Composition—regular	0.283 ***	0.063	−1.673 ***	(−2.410, −0.935)
Team Composition—advanced	0.386 ***	0.064	−2.282 ***	(−3.202, −1.362)
Workload—20~39 patients	−0.030	0.055	0.179	(−0.468, 0.826)
Workload—40~59 patients	−0.222 ***	0.061	1.310 ***	(0.605, 2.016)
Visit pattern—telephone + outpatient visits	0.064	0.054	−0.376	(−1.012, 0.259)
Visit pattern—telephone + home visits	0.002	0.051	−0.014	(−0.608, 0.580)
Incentive—no rewards for doing well, but penalties for not doing well	0.082	0.055	−0.487	(−1.119, 0.145)
Incentive—rewards for doing well, and penalties for not doing well	0.152 *	0.064	−0.899 *	(−1.612, −0.186)
Adherence of patients—good	0.244 ***	0.065	−1.442 ***	(−2.327, −0.557)
Adherence of patients—very good	0.128 *	0.056	−0.759 *	(−1.446, −0.072)
Number of observations	10560			
Log likelihood	−3563.99			
Wald Chi2(df)	190.68 (12)			
Prob > Chi2	0.000			

Notes: ① ^#^ reference categories are the 1st levels of each attribute. ② * 0.01 ≤ *p* ≤ 0.05; *** *p* < 0.001.

**Table 4 ijerph-18-08317-t004:** Analysis with interaction terms.

Variables ^#^	Coefficient Means	Robust S.E.
Constant	−0.156	0.129
Payment	0.170 ***	0.018
Team Composition—regular	0.541 ***	0.113
×female	−0.365 **	0.132
Team Composition—advanced	0.329 ***	0.069
×bachelor or higher	0.515 **	0.177
Workload—20~39 patients	−0.029	0.056
Workload—40~59 patients	−0.234 ***	0.062
Visit pattern—telephone + outpatient visits	0.070	0.055
Visit pattern—telephone + home visits	0.234 *	0.116
×bachelor or higher	0.402*	0.175
Incentive—no rewards for doing well, but penalties for not doing well	0.076	0.056
Incentive—rewards for doing well, and penalties for not doing well	0.253 **	0.081
×technical school	−0.297 *	0.141
Adherence of patients—good	0.435 ***	0.111
×work more than 5 years	−0.288 *	0.136
Adherence of patients—very good	0.130 *	0.056
Number of observations	10560	
Log likelihood	−3524.45	
Wald Chi2(df)	220.85(18)	
Prob > Chi2	0.000	

Notes: ① ^#^ reference categories are the 1st levels of each attribute. ② * 0.01 ≤ *p* ≤ 0.05; ** 0.001 ≤ *p* < 0.01; *** *p* < 0.001.

## Data Availability

The datasets used and analyzed during the current study are available from the corresponding author on reasonable request.

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
