# Peer review of "Eliciting Preferences of Providers in Primary Care Settings for Post Hospital Discharge Patient Follow-Up"

_ijerph, 2021, doi:10.3390/ijerph18168317_

Round 1

Reviewer 1 Report

This is a well written paper with scientific data analysis to address five attributes: team composition, workload, visit pattern, adherence of patients, and incentive mechanism to evaluate primary care providers’ preferences for delivering post-charge follow-up care for patients with chronic diseases. We could have the results from physicians, nurses, and other medical co-workers with interactional data analysis.

Tables are well organized and elaborated to address the variables in different levels.

However, I am curious about the duration for data collection in this study, which maybe involved with the time of pre-COVID and under-COIVD (18months so far). I believe there would be something different in the two much different scenarios, especially for the result interpretation. It should be well-analyzed in the discussion or well-explained in the material and method to segment the two different timing. By the way, it would be more interesting to tell all the audiences about the topic in the circumstance of COVID.

Please add the above information. Otherwise, this is a good paper.

Thank you.

Author Response

Thanks for the comments.

Our survey was conducted in November 2020 when COVID-19 was well under-control in China, so it isn’t appropriate to compare the two scenarios as pre-COVID and under-COIVD in our study.

We do agree with your suggestion about considering the topic of COVID-19, so we added “In fact, the COVID-19 pandemic has also highlighted the importance of proactive multidisciplinary team-based care.” in our discussion (page 8).

Reviewer 2 Report

The article: “Eliciting preferences for provides in primary care settings for post hospital discharge patient follow-up” was of interest given the topic (providers’ preferences for patient follow-up post discharge), the analytic methods (mixed qualitative and quantitative) and the design (discrete choice experiment). 

This draft needs attention to purpose, reproducibility, and utility.

The authors seek to add to the knowledge of  motivates providers to provide follow-up to patients once discharged from hospital care. Within the discussion section, the authors principal finding is  to“(…) provide evidence for designing an implementation plan for post discharge follow-up in primary practice” (lines 256-257, page 8).  They combine primary data collection with qualitative interviews to conclude that the solution is multidisciplinary teams and a focus on motivating certain providers. Such conclusions are not substantiated by the data nor the analysis. I encourage the authors to align their purpose, methods, and results. These results, while interesting, are not generalizable. Thus, be careful not to extrapolate beyond the data when interpreting results. With its focus on discrete choice and economic trade-offs, the centrality of income must be treated carefully by different income groups. People’s willingness to pay is in large part a function of their household incomes. 1

The methods section contains too much detail of statistical theory and not enough information for reproducibility. Remember your audience. The appeal and reach of the IJERP means you must tailor your content to the intended audience. Consider replacing most of the content within section 2.2 DCE Design (pages 3 & 4) with rationale for why the models were focused on team composition, workload, visit patterns, incentive, and adherence of patients; and not confounders such as provider type (e.g. physicians, nurses etc.). While I agree with the authors that the primary contribution of this paper is the methodological innovation, the authors’ interest in the five attributes could benefit from a conceptual model (directed acyclic graph, or DAG) that connect attributes to outcomes and known confounders.

The value of the two exhibits (modeling output displayed in Table 3 and Table 4) are difficult to decipher given the broad heterogeneity among survey respondents (Table 2). Instead of providing coefficients (Tables 3 and 4), other statistical methods may yield more illuminating results. For example, please consider cluster analyses to identify groups with similar perspectives as an alternative to displaying coefficients and interaction terms. The key is to make your findings actionable.

As a fellow-English as a Second Language (ESL) scientist, please consider engaging a professional copy-editor. I’ve found such professionals invaluable in correcting syntax, clarifying sentence construction, and making the article more accessible to a broader audience.

1 Wang, Y., & Zhang, L. (2019). Status of public–private partnership recognition and willingness to pay for private health care in China. The International Journal of health planning and management, 34(2), e1188-e1199.

Reviewer 3 Report

Dear Authors

I read with interest your paper. The study aims to assess primary care providers' preferences for delivering post-charge follow-up care for patients with chronic diseases. The topic is fascinating.

Introduction: This section is clear and well-written. You clearly described the aim of the study.

Methods: You clearly described the methods applied for data analysis in the present study. On page 3, line 125, you stated that "A full factorial design produced 36=729 scenarios". What did you mean? I think that "=" was a mistake. 

Results: This section is clear and well-written.

Discussion: You reported a personal point-of-view for a nurse about the choosing of an advanced follow-up team. The authors should remove this part and discuss the results, avoiding reporting unique and unexplained point-of-view.

In conclusion, my final decision was that the paper required minor revision.

Author Response

1. Methods: You clearly described the methods applied for data analysis in the present study. On page 3, line 125, you stated that "A full factorial design produced 36=729 scenarios". What did you mean? I think that "=" was a mistake.

We would like to thank the reviewer’s detailed correction. This was a format error which should be ”36=729”. We have corrected it in the manuscript.

2. Discussion: You reported a personal point-of-view for a nurse about the choosing of an advanced follow-up team. The authors should remove this part and discuss the results, avoiding reporting unique and unexplained point-of-view.

We realised the inappropriateness in reporting a personal point-of-view to explain the results of data analysis. Adjustments have been made in the discussion section (page 8).

Round 2

Reviewer 1 Report

Dear authors,

Realizing the condition is since November, 2020, COVID is well in China. Accept after you add information here.

Author Response

Thanks for the comment.

We have added the information in the part "2.1 study setting and sampling".

"The survey was conducted in November 2020, after China managed to control COVID19."